# Review of Urban Flood Resilience: Insights from Scientometric and Systematic Analysis

**DOI:** 10.3390/ijerph19148837

**Published:** 2022-07-21

**Authors:** Meiyan Gao, Zongmin Wang, Haibo Yang

**Affiliations:** 1Yellow River Laboratory, Zhengzhou University, Zhengzhou 450001, China; gmeiyan123@163.com (M.G.); zmwang@zzu.edu.cn (Z.W.); 2School of Water Conservancy Engineering, Zhengzhou University, Zhengzhou 450001, China

**Keywords:** urban flood, resilience, climate change, bibliometric, cite space

## Abstract

In recent decades, climate change is exacerbating meteorological disasters around the world, causing more serious urban flood disaster losses. Many solutions in related research have been proposed to enhance urban adaptation to climate change, including urban flooding simulations, risk reduction and urban flood-resistance capacity. In this paper we provide a thorough review of urban flood-resilience using scientometric and systematic analysis. Using Cite Space and VOS viewer, we conducted a scientometric analysis to quantitively analyze related papers from the Web of Science Core Collection from 1999 to 2021 with urban flood resilience as the keyword. We systematically summarize the relationship of urban flood resilience, including co-citation analysis of keywords, authors, research institutions, countries, and research trends. The scientometric results show that four stages can be distinguished to indicate the evolution of different keywords in urban flood management from 1999, and urban flood resilience has become a research hotspot with a significant increase globally since 2015. The research methods and progress of urban flood resilience in these four related fields are systematically analyzed, including climate change, urban planning, urban system adaptation and urban flood-simulation models. Climate change has been of high interest in urban flood-resilience research. Urban planning and the adaptation of urban systems differ in terms of human involvement and local policies, while more dynamic factors need to be jointly described. Models are mostly evaluated with indicators, and comprehensive resilience studies based on traditional models are needed for multi-level and higher performance models. Consequently, more studies about urban flood resilience based on local policies and dynamics within global urban areas combined with fine simulation are needed in the future, improving the concept of resilience as applied to urban flood-risk-management and assessment.

## 1. Introduction

As the Intergovernmental Panel on Climate Change (IPCC) points out in its Sixth Assessment Report (AR6), humans will face more extreme events. As global temperatures continue to rise, early warning of extreme hazards is needed. The emergence of an extreme global climate will cause more meteorological disasters [1]. At present, urban climate change has become an important challenge faced by all countries in the world [2]. Due to the rapid development and expansion of cities, their highly concentrated population and high intensity of economic activities, the degree of exposure to natural disaster risks is constantly increasing. Now, new composite urban disasters, such as typhoons, urban waterlogging and urban heat islands, have caused the loss of urban meteorological disasters. Among them, urban flooding is one of the meteorological disasters that all regions may encounter, and it is closely and complexly related to climate, urban planning, drainage and human activities [3]. Urban areas are different from rural areas. In urban areas, the causes of flood risk are more complex, which is related to a large number of land types with different uses, concentrated buildings and a large number of projects in cities [4,5]. In addition, urban flood disasters also bring huge economic losses to the dense economy in the city, and affect the safety of life of all urban residents. Urban flood resilience was defined in the EPSRC project as a city’s capacity to maintain future flood risk at tolerable levels by preventing deaths and injuries, minimizing damage and disruption during floods, and recovering quickly afterwards, while ensuring social equity and protecting the city’s cultural identity and economic vitality. In recent years, more countries and regions have been affected by urban floods. Since 1989, more than 4000 major floods have occurred worldwide. China, India, the United States and Indonesia have the largest number of flood disasters, with a total of more than 1200 occurrences. Practically, urban rainfall can directly regulate air humidity in cities. Urban flooding has both advantages and disadvantages. It replenishes surface water in urban rivers and wetlands [6], increases the amount of water stored in reservoirs [7], and has a positive effect on groundwater replenishment in urban areas [8]. The urban ecosystem can also be improved after being replenished with water [9]. At the same time, floods carry some of the sediment flow and replenish the nutrients of the land downstream. On the contrary, the disadvantages of urban flooding become particularly evident when rainfall exceeds the carrying capacity of the urban system. On the basis of the huge economic loss caused by urban flood disasters, it also affects the safety of human life. Major urban floods include the Bangkok flood in Thailand in 2011 [10], the “7.21” Beijing rainstorm in 2012 [11], and the “7.20” extreme rainstorm in Zhengzhou in in 2021. These rainstorms have caused huge casualties and property losses in urban areas or urban agglomerations, attracting widespread attention from all of society. According to the investigation report on the “7–20” extraordinarily heavy rainstorm disaster in Zhengzhou, Henan Province, published by the Disaster Investigation Group of the Chinese State Council, the urban flood disaster in Zhengzhou city caused direct economic losses of 7.9 billion USD, damaged 400,000 vehicles and killed 292 people. Urban flood disasters, as an unstable urban disturbance factor [12], may lead to disordering the operation of an urban system. With the consequences of urban flooding, such as loss of the life and property of residents and the disruption of social development, the ability of cities to cope with climate disaster crises needs to be improved. In the context of global climate change, more and more extreme precipitation will occur, and research on the resilience of cities against flood disasters caused by extreme precipitation has thus attracted people’s attention in order to cope with the increasing frequency and intensity of urban climate disasters caused by climate change, and to ensure the ability of urban areas to cope with risks. 

For coping with the urban risks caused by climate change, the concept of urban resilience has been put forward in the international community [13]. In 2012, UN-Habitat proposed the urban resilience study, which aims to adapt cities to climate disasters while helping them improve their ability to cope with external shocks and disasters. Cities are characterized by dense population and concentrated industries. Thus, the ability of cities to resist and recover in the face of floods plays a very important role in safeguarding economic development and the safety of life and property [14,15]. With the increasing degree of urbanization, climate change caused by the corresponding heat island effect will also become an important development in reducing urban losses. The term ‘resilience’ is now used in a wide range of journals, institutions and fields. According to the search results of Web of Science with ‘urban flood resilience’ as the keyword, resilience can be applied to the coping capacity and resilience of cities to flood. Moreover, it has been studied in 152 fields such as psychology, environmental science, computer science and engineering, for example, power-sensitive resilience [16], critical infrastructure resilience [17], mechanism and managerial resilience [18] and cyber-physical resilience [19]. More than 70,000 articles have been published in the past five years. At present, there are relevant studies on urban resilience. Urban resilience refers to the adaptive capacity of all urban systems, and urban flood resilience is equivalent to the adaptive capacity to cope with floods caused by climate change in urban resilience research [20]. Flood resilience can greatly reduce the adverse effects of extreme weather [21]. When a city endures a flood disaster, a wide range of its influencing factors are designed. At the same time, cities are always affected by human activities and the selection of urban planning and design standards for pipeline networks. Urban flood resilience depends on the standards followed by designers and decision makers considering factors such as the number of urban permanent residents, local ecological environment, socio-economic development, flood control and drainage capabilities.

Modern methods of dissertation mining and analysis complete the field of urban flood-resilience research and can be of great help to scientific research within a field [22,23,24,25]. Scientifically based econometric analysis often focuses on data analysis and multiple relationship graphs to illustrate relationships in the literature, while qualitative analysis focuses more on verbal text to illustrate and explore research methods, differences and gaps in a vast literature. The combination of these two approaches allows for better analysis of research trends on research topics and more accurate and quicker analysis of research subjects. Previous quantitative analysis literature containing ‘flood’ as a keyword contains such aspects as flood risk perception [26,27], flood risk assessment [28] and disaster management [29]. At the same time, there are also researchers who conduct systematic literature mining based on more targeted research topics, such as sponge city [30], stormwater management [31] and urban floods [32]. Scientometric analysis can apply various literature analysis tools such as VOSviewer [25], Citespace [33], BICOMB [34,35] and Histcite [36]. Targeted research can be conducted by adopting analysis tools and methods that are more specific to the characteristics and trends of the research field according to the differences in the central terms of the research. Thus, trends and hotspots can be analyzed for studies with higher relevance and closer to the keywords. The use of scientometric research is very helpful for the analysis of clear research developments in urban flood resilience.

In recent years, the number of studies on urban flood resilience has gradually increased. Cities are complex systems that include economic, social, ecological and human dimensions, and impervious and impermeable surfaces exist at cross purposes. Urban flooding is an integrated meteorological hazard and includes numerous influencing factors within the city that have an impact on the study of urban flood resilience. Urban flood resilience is part of traditional urban resilience research and a more specialized area of resilience research based on urban response to disasters. Although scholars have conducted several studies in urban flood resilience, normative and scientific econometric analysis is still an important part of literature research. In this paper, we apply bibliometric methods to sort out the hotspots and trends of urban flood resilience research triggered by heavy rainfall and floods from the perspective of quantitative analysis of objective literature. On the basis of analyzing the current status of urban flood-resilience research, we combine and discuss the research hotspots of urban flood resilience, hoping to help subsequent research on urban flood resilience.

## 2. Scientometric Analysis

### 2.1. Materials and Methods

This review follows a process figure (Figure 1) for a scientometric and systematic analysis of the literature that relates to urban flood resilience. In order to clarify the current research status and development trends of urban flood resilience, Cite Space software (Chaomei Chen, Philadelphia, PA, USA) was used for keyword frequency statistics and high-frequency keyword screening analysis. VOS Viewer software (Nees Jan van Eck and Ludo Waltman, Leidon, The Netherlands) was used for social network analysis and hierarchical clustering analysis, which finally presents the research changes of urban flooding in a quantitative and dynamic way. Research hotspots in different stages are compared, future trends are predicted according to keywords, and future research directions are explored. Cite Space is an information visualization software that can analyze the characteristics of the literature in various research fields. It includes analysis related to keywords, authors, cited studies, countries, research institutions and journals. It provides visual forms of keyword clustering, cooperative networks, co-cited networks, literature sources, regional distribution and so on.

In this study, Web of Science (WOS) was selected as the literature search engine, and the data source was the Web of Science Core Collection. The database is authoritative in terms of the number of journals, the number of publications, research field and time span, so it is sufficient to carry out literature visualization analysis on the basis of this database [37]. ‘Urban flood resilience’ was used as a keyword for retrieval. The literature search results were obtained by searching for ‘urban flood resilience’ as a keyword and employing a restricted literature publication time to filter the literature. The retrieval time was determined to be from 1 January 1999 to 31 December 2021. The repeated calibration of the search words and comparison of the search formula for multiple subject retrieval method is adopted. The literature search formula was (literature language = English, literature type = Article AND Review, time span = 1999–2021). A total of 1420 records were retrieved. A total of 1416 valid data were obtained based on the removal of duplicate and irrelevant documents. Analysis was carried out, ranging from keyword analysis to journal source analysis, research institution analysis, country analysis, author co-citation analysis and so on. The keywords were analyzed in stages to perform hotspot analysis and predict the development trends of keyword changes in the literature according to the timeline.

### 2.2. Keywords Analysis

Taking each year as a time slice, the literature on the design of “urban flood resilience” from 1999 to 2021 was analyzed. VOS viewer software was used to conduct preliminary keyword analysis on the selected literature. A total of 146 keywords were analyzed and 497 keywords were connected. The top 20 keywords were selected to form a knowledge network, and keyword analysis was conducted (Figure 2). Keyword analysis shows the keywords co-cited with urban flood resilience, followed by the number of keywords appearing as literature keywords in parentheses, including: resilience (554), climate change (472), vulnerability (314), management (215), adaptation (212), framework (128), flood (118), flood risk (114), model (105), city (104) and impact (102). Hot keywords are useful for subsequent hot research directions and research planning on urban flood resilience. The keyword ‘resilience’ is the most frequently cited. This shows that our literature search and analysis was effective this time, and the literature was distributed closely around the term resilience. Additionally, this indicates that studies on urban flood resilience focus on climate change, vulnerability, management, risk, impacts, etc. At the same time, the keywords involve multiple fields, activity subjects and management subjects in the city, so the research related to urban flood resilience still needs to be studied in many aspects. Therefore, studies on urban flood resilience tend to focus on urban management and disaster-bearing capacity, but lack investigation of their practical application.

The figures show the temporal change trends of hot keywords from 1999 to 2021 (Figure 3). The number of studies on urban flood resilience has been increasing since 2010, and the number of literature items related to urban flood resilience has increased significantly since 2016, when urban flood resilience became a hot research study topic. From 1999 to 2021, the study was divided into four stages showing the evolution of different keywords in urban flood resilience, 1999–2005, 2006–2010, 2011–2015 and 2016–2021. The completion of the literature co-citation analysis revealed little literature from 1999 to 2005 and little correlation within it. In the literature between 2006 and 2010, studies related to climate change and flooding began to emerge. From 2011 to 2015, applied research articles with water management, communication, management and flood risk management as keywords became hotspots. From the perspective of keywords, their core lies in ‘management,’ which is more inclined to be targeted at research phenomena. The impact of extreme weather in cities due to climate change has received attention. There has been a substantial increase in research closely related to climate change, including research into sponge cities, climate adaptation, urban flood resilience, urban resilience and urban climate adaptation. In addition, it can be seen that, although the keyword relationship is relatively intensive (Figure 2), the articles classified by keywords in the other three stages are relatively clear. Between 2016 and 2021, literature with the term ‘urban flooding’ as the central term is more relevant. In Figure 3, it is shown to be dense with few dispersed nodes. The keywords are all related to the city’s ability to withstand climate change and its resilience, with climate change as the core.

### 2.3. Author and Institution Analysis

Figure 4 shows that an author-co-citation and national-cooperative relationship between authors can be identified. The connection between two authors represents the existence of cooperation between them, and the distance between authors is what is called the total relationship. In the author co-citation network, there are 812 nodes. In author co-citation analysis, only authors with more than 25 citations are shown. It can be seen that articles published by closely related authors are mostly distributed after 2010. Among them, the author of the most cited literature is Meerow S (frequency = 82), followed by Fletcher TD (frequency = 36), Kotzee I (frequency = 33), Bertilsson L (frequency = 30), Mugume SN (frequency = 30). The authors are from different countries. The number of citations of their paper can fully indicate that their paper is worth learning and also has important reference value in the research field of urban flood resilience. The threshold is set to 50. The first source country of the literature is the United States, followed by the United Kingdom, China, the Netherlands, Germany, Australia, and Italy. The three countries that contributed the most have published 354, 221 and 161 articles, accounting for 25%, 15.6% and 11.4% of the total number of published articles using ‘urban flood resilience’ as a keyword.

As for the cooperation network of research institutions on urban flood resilience, it can be clearly seen that the University of Exeter conducts the most research on this topic (Figure 5). Most research institutions have some cooperative relationship with them. From the network analysis of the document-source countries, it can be seen that the research on urban flood resilience has been carried out by many departments from different countries and regions, and there is a certain degree of relevance. The top ten relevant research organizations with the highest number of publications in the literature were selected, ranked and cited as shown in Table 1. The large difference between the number of literature items and citations indicates that the literature from these research organizations is mostly valid and highly cited.

### 2.4. Sources of Literature Analysis

Among these studies, there are attempts based on a variety of frameworks, methods and indicators, but in general, the research on urban flood resilience still lacks validated measurement standards [38]. As the country with the largest number of publications related to urban flood resilience, the United States has tried to propose some theories that have been applied traditionally or not yet been verified in a large number of applications. The related methods used in these studies are the fuzzy Delphi method (FDM) and the analytic network process (ANP) [39] geographic information system (GIS) [40], cluster analysis method and bivariate correlation method [41], etc. At the same time, NAT HAZARDS, GLOBAL ENVIRON CHANG, ENVIRON SCI POLICY and J HYDROL were co-cited and found to be the most relevant journals with high correlation (Figure 6).

### 2.5. Research Hotspot

By being analyzed by Cite Space software, relevant studies have increased significantly since 2017, showing a trend of rapid increase, and have maintained annual growth since then (Figure 7). Simultaneously, after further clustering analysis of keyword networks, 10 clusters were obtained, among which 8 clusters were considered suitable for clustering (Sihouette > 0.7, and modularity was between 0.4 and 0.8) [42]. The clustering results show that the study of urban flood resilience covers a wide range of fields, and there are certain correlations within the same subject. The clusters with the strongest similarity and the largest number of tube tests in the cluster are: managing urban drainage system, hydro-meteorological risk assessment method, urban planning, urban drainage system, case study, systematic review, riverbank region and urban resilience. Currently, applied research on urban flood resilience accounts for a large part of this, and it has grown rapidly in recent years. However, there are still gaps in the pure theoretical research on urban flood resilience and even resilience on its own.

## 3. Systematic Analysis

### 3.1. Urban Flood Resilience and Climate Change Adaptation

#### 3.1.1. Urban Flood Resilience and Climate Change

In the previous keyword-phase analysis, climate change was analyzed as one of the core keywords in three-phase analysis graphs since 2006. Climate change is in the center of the keyword analysis in all of them. Thus, climate change is one of the most influential factors among the keywords related to the study of urban flood resilience, which is also affecting urban runoff as well as natural river basins [43]. In the face of global climate-change challenges, countries need to explore new technologies and methods to make cities more resilient, resilient and sustainable in coping with floods. Urban resilience is a comprehensive urban response to various disasters and resilience, and urban flood resilience is a more detailed part of urban flood response; urban resilience and urban flood resilience are largely interactive. Urban flood resilience is a more targeted study. The increasing number of studies on urban flood resilience is closely related to climate change. The extreme weather caused by climate change will have an irreversible impact on the built-up areas of cities, and even cause great damage to the area around cities after floods. Therefore, it is important to clarify the development and trends of research issues linked to climate-change-related urban flood resilience.

The keywords with the highest collinear frequency in relation to urban flood resilience are resilience, vulnerability, climate change, climate-change, adaptation, management, risk, etc. Among these, climate change and resilience are the key points. Climate change in different climates, countries and regions, and the adaptation degree of cities to climate change have also aroused research attention [44]. The countries in the study show that many countries and regions, including Europe, China, Australia and France, are very concerned about climate change and urban flood resilience. In 2013, Wachinger G discussed preventive behavior resulting from personal experience, culture and personal factors in climate change events such as floods [45]. In 2016, Cutter SL made a core set of attributes, assets and proxy measures based on the analysis of 27 resiliency assessment tools and indicators [46]. In 2018, Crncevic T provided a comprehensive insight analysis of the relations between climate change and DRR, with special reference to planning policy. In the same year, recommendations on the need to improve the resilience of Serbian communities to disaster risk during floods were presented in the light of disasters caused by climate change [47]. Tauhid FA adopted the multi-case study method to study the effects of urban floods on green infrastructure (GI) and the urban poor based on the current situation of climate change and rapid urbanization. The cooperation and full participation of urban stakeholders are key issues linked to the impact of climate-related floods [48]. The ability of cities to cope with climate change also depends in part on government policy decisions regarding urban flooding. Successful urban strategies include green roofs, rainwater tanks, permeable urban surfaces, surface rainwater management and local stormwater disposal [49]. Generally speaking, urban flood resilience is also the embodiment of urban disaster resistance [50]. Urban vulnerability and vulnerability analysis are aimed at studying resilience to urban floods and climate change. Urban flood resilience is reflected in the ability to cope with extreme precipitation. So far, with the increasing number of cities and the growing urban economy, there has been a corresponding increase in related research in various countries and research institutions. In the current climate conditions, the study of urban flood resilience is deepening, which also reflects the continuous improvement of our urban flood-response ability.

#### 3.1.2. Urban Flood Resilience and Hydro-Meteorological Risk Assessment

According to our keyword cluster analysis, hydro-meteorological risk assessment is a cluster, which is closely related to urban flood and urban flood resilience. Therefore, it is classified and discussed under climate-change adaptation. The emergence of urban flood climate factors is important. With respect to the city of Wuhan in Hubei province in China, the geographical factors of the city determine that the city suffers from the impact of floods in the rainy season, so in the weather risk assessment of these types of city environment [51], the original urban planning must determine the ability of the city under the flood. Cities are asset-intensive and population-concentrated, so timely, effective and meaningful hydrometeorological risk information is important for rapid response, personnel coordination and climate monitoring in urban flood situations. In urban flood risk assessment, hydrometeorological risk assessment (HMR) and other different methods, including open data [52], multi-source remote-sensing precipitation products [53], and GIS and other software, can be used. In addition, principal component analysis [54], the extremum theorem, functional data analysis and other computational [55], mathematical and statistical methods [24], along with the Ornstein–Uhlenbeck process and meta-analyses [56], can be used to study urban flood resilience. In 2014, Revi A et al. considered the adaptation capacity of cities in the world to climate change and discussed the risk level of factors driving short-term and long-term climate impacts [57]. In 2017, Demarchi used GIS to integrate urban information at different levels, such as risk, exposed assets or vulnerability types. Simultaneously, satellite precipitation data were combined with geospatial reference data to improve urban flood resistance [58]. In 2018, Djalante R proposed that in urban risk assessment, communities are equally important to the progress of research on disasters and climate change in Indonesia, including the economic and social impacts of disasters and climate change on vulnerable regions [59].

Likewise, in the process of hydro-meteorological risk assessment in urban areas, the index system of risk assessment is an important method of resilience assessment when constructing the main body of the urban flood-resilience index. Cutter has built a six-dimensional and 30-point community toughness index system, including a system of ecological, social, economic, infrastructure and community [60]. On this basis, some scholars have improved the index and evaluation area. For example, Ruan, JE integrates urban economy, society and ecology as elements of the evaluation model [54,61] in the evaluation of urban flood resilience by using index system. At present, methods include the fuzzy comprehensive evaluation method (PCA), principal component analysis and analytic hierarchy process [62], factor analysis [63], TOPSIS [64] and other methods. The weights of toughness indexes were ranked to reflect the strengths and weaknesses of each evaluation object or index.

### 3.2. Urban Flood Resilience and Urban Planning

#### 3.2.1. Urban Flood Resilience and Urban Planning and Design

In the cluster analysis, the terms related to urban planning include urban management, drainage, governance, etc. Urban planning and design are an important basis for studying urban flood resilience. Water management in urban planning in cities is influenced by many social, economic and environmental factors [65]. The size, population and property of cities are also increasing with continuous development. The dense distribution of buildings, population and property in a city also affects the climate of the urban area and the drainage capacity of the city. As for the study of urban flood resilience, the original design and planning of the city is the most basic research framework of the city. The city is accompanied by intensive human activities, which are important influences in the study of urban flood resilience. In addition, the continuous development of urbanization has changed the original hydrological environment, and the pervious area in the city has gradually decreased, while the impervious area has increased, and the urban buildings have also been increasing [66]. It is also very important to discuss urban flood resilience from the perspective of urban planning and design. Through urban planning and urban design, the resilience and the vulnerability of the city can be improved. Proper urban planning can make cities more resilient to floods while achieving sustainable development. In the process of urban flood-resilience planning, a city’s flood resilience within a fixed period of time is evaluated [67], and a resilience improvement scheme is proposed for the old urban infrastructure [50]. In 2014, Demuzere M et al. made proposals for the planning and environmental management of urban areas based on the function of green space to serve in climate-change mitigation and adaptation on three spatial scales of city, community and site-specific [68]. Taking the Mekong Delta in Vietnam as an example, Liao proposed a flood adaptation model that allows floods to enter cities. He also mentioned three urban planning and design rules: anticipate and adapt to floods, incorporate ecological processes of floods and reveal flood dynamics [69]. In 2019, Bertilsson L modeled and spatialized flood resilience through a spatialized urban flood-resilience index (S-FRESI), measuring and visualizing flood-resilience changes obtained by different flood control measures in response to the phenomenon of continuous urbanization growth using S-FRESI [70]. In 2020, O’Donnell E expounded three research themes of resident urban flood resilience (UFR) [71]. Meanwhile, flood risk, wastewater and rainstorm management should be re-envisaged and transformed to ensure satisfactory service delivery under flood, normal and drought conditions, to enhance and extend the useful lives of ageing grey assets by supplementing them with multi-functional blue-green infrastructure [72]. Therefore, urban flood resilience is significantly related to urban structure, urban infrastructure, layout, the lifetime cycles of building [73] and future urban development plans.

#### 3.2.2. Urban Flood Resilience and Urban Drainage System

Rapid urbanization has increased the number of impervious surfaces and changed the flow route of flooding in the city. The study of urban drainage systems is closely related to the resilience to urban floods. Learning how to mitigate the consequences of flooding and coexist with floods plays a very important and positive theoretical and practical role in urban flood response [74,75]. It is necessary to evaluate the flood-resistance ability of urban drainage systems during the planning and design stage. In 2016, Sorensen J proposed the concept of improving urban flood resilience based on the resilience theory. A synergy between the ability to handle stormwater runoff and improving the experience and functional quality of the urban environment was also achieved [76]. In 2021, Wang MM proposed a benchmark method for system-resilience assessment based on “do nothing”—that is, obtaining the number of inundation nodes used to calculate the average flood duration without doing anything—and explored improving the system resilience assessment of urban floods [77]. In view of the grate-blocking phenomenon and more intense storm events caused by climate change under critical conditions, Russo B put forward that urban drainage-system design is mostly deficient in its underground network, and the surface drainage system plays a positive role in urban flood resilience under flood conditions [78]. Mattos TS evaluates the resilience of stormwater drainage systems in the form of low-impact development (LID), and evaluates flood-resistance capacity through runoff peak value and a drainage resilience index [79]. Hou JM proposed a linear reservoir method to characterize the drainage capacity of the pipe network, while a high-resolution digital elevation model (DEM) was used to characterize the microscopic characteristics of the city [80].

Resilience theory is the predecessor of urban flood-resilience research, and this method can also be applied to it. The methods used in the existing literature include the entropy method [66,81], integrated climatological-hydrodynamic method [82] and black box evaluation method [83], as well as the new evaluation index system and the proposed model based on this. There are also studies based on the BRIC model proposed by Carter [84]. An urban drainage system can also be evaluated by spatial analysis, and urban flood resilience can be determined on the basis of this evaluation. On the other hand, the optimal allocation of urban drainage systems for flood discharge in different regions also has a positive impact on flood-control management and flood-disaster reduction. The reasonable allocation of flood discharge can reduce the pressure on the city to deal with flooding in extreme weather [85,86]. There are also proposals to combine toughness theory with machine learning to respond. During a disaster, urban factors such as population, industry, urban assets and the drainage pipe network are not only the basis of ensuring the resilience of the city in flood, but also the factors of loss in the process of urban disaster. Urban flood resilience represents the balance between and adaptation of city and flood. It is also the ability and speed of a city to recover from a flood disaster to its equilibrium state.

### 3.3. Urban Flood Resilience and Urban System Adaptation

The purpose of urban flood resilience is to enhance the capacity of urban flood control and disaster reduction, which must be based on risk assessment to evaluate the adaptability of urban response to disasters [87,88]. The definition of urban flood resilience is now variously defined, but urban flood resilience also shares the same capabilities as urban resilience; stabilization, resilience, and adaptive capacity. The differences that exist in different definitions of the urban flood-resilience stem from the different biases towards these three capabilities, including but not limited to the ability of cities to cope with floods based on urban resilience, disaster resilience and urban risk assessment [89]. The adjustment of the weighting of social, economic, and ecological aspects is flexible according to the research area [90,91]. For example, the definitions of different fields such as roads [92], ecology and economic loss [93] are adjusted according to different research subjects. Moreover, different definitions of urban flood resilience can be made for different subjects, but there is a lack of definition from subjects. The research on urban flood resilience originated from urban flood disasters caused by rainstorms, so it is closely related to disaster analysis. When dealing with floods, the analysis and evaluation of a city’s bearing capacity and the damage capacity of flood disasters provide optimization methods for the evaluation of urban flood resilience [94]. The research focus is on flood-disaster analysis to allow the existing state of urban flood resilience to be improved. In 2015, Golz S used an analytic hierarchy process (AHP) to quantify and compare the physical flood vulnerability of building structures [95]. In 2019, Wang YT proposed a toughness measurement method based on grid cells to measure the urban surface flood toughness according to the urban scale, and simulated the surface flooding in Dalian using the cellular-automata-based model CADDIES [96]. In 2021, Xu WP used a model combining the interpretive-structure and network-analysis methods (ISM-ANP) to evaluate and analyze the selected evaluation indicators. The ranking of key indicators of flood resistance for Wuhan, Nanjing and Hefei in China is helpful to understand urban flood resilience [97]. Park K proposed a method to reduce urban flood damage and improve urban resilience, taking land use and building characteristics as evaluation indexes for urban flood-vulnerability analysis. Disaster resistance can be improved through proper land-use planning and urban planning-facility selection [98]. Understanding the changing laws of urban flooding is of great significance to reduce and adapt to flood risk [35]. As the urban environment changes, the ecological footprint (EF) metric can also be used to evaluate the sustainable development of the original urban system [99], to increase the resilience of urban systems to flooding and improve flood protection through urban design in collaboration with researchers, including improving infrastructure levels and increasing hollow areas and drainage capacity [100,101,102]. These facilities, from the perspective of a city adapting to this strategy, can be part of the development of an urban flood-toughness plan.

### 3.4. Urban Flood Resilience and Flood Simulation Models

As an important method for studying floods, models also have a very important role in the study of urban resilience caused by floods [102]. According to the results of keyword analysis in Section 3.1, ‘models,’ as one of the keywords, is relevant for the study of resilience and vulnerability of cities to floods, as well as natural phenomena and human measures closely related to urban development, such as climate change and urban management. This model of disaster-risk research adopts many system-evaluation methods in an urban research system, according to existing urban flood-resilience-related studies involved in the quantification of specific values according to the model.

Most human-related natural hazards occur in cities, and there is now an overlap between the research literature on urban resilience and disaster-related literature. Flood resilience research with urban systems as the object of study is an important framework for the quantitative assessment of the resilience of cities to disasters such as floods [103]. While disaster resilience is the resilience of a system in the face of widespread hazards, urban flood resilience is the scope of resilience studies delineated for extreme precipitation [104]. Urban flood-disaster resilience is one of the branches of research on vulnerability and resilience for floods applied to urban areas. The progress of research methods for disaster-risk-resilience assessment is a reference for flood resilience research. The use of models for urban resilience assessment generally includes the application of the following types of models: commonly used disaster-risk-assessment models, models for the assessment of flooding hazards and system-performance-evaluation models. Common disaster-risk assessments used in flood resilience assessment include CDRI [105,106] and TOPSIS [64,107]. The resilience-assessment models with applied indicator systems are more frequently applied. Different assessment systems use different assessment indicators selected for different research directions, and a frequently used calculation method is the assignment of weights to calculate the assessment-index method. The BRIC model was applied to 3108 counties in the United States for disaster-resilience assessment [108]. The LDRI (the localized disaster-resilience index) model was applied to coastal communities in the Philippines for disaster-resilience assessment [109]. The REDI model assessed community resilience based on high-resolution urban data [110]. Some researchers combined various models and tools to evaluate the vulnerability, resilience and vulnerability of flood research objects in disasters, and conducted a sponge-city evaluation based on the SCS and flood models constructed by GIS [111,112]. The 2D flood inundation model (FloodMap-Inertiamo) was applied to coastal areas to predict the inundation of floods, based on the local seal-level rise and long-term ground circulation, to evaluate the exposure and vulnerability of population and property in the floodplain [113]. There are also studies on the Learning from Floods (LFF) model to compare flood resilience in two environments [114]. In addition to traditional models, some scholars also predict flood risk based on high-performance [115,116] and large-scale flood models [52], in order to propose development strategies for urban flood resilience.

In addition to the integrated assessment, there are also the development of economically targeted property flood-resilience in flood-risk management (targeted PFR) models, the index-based spatialized urban flood resilience index (S-FRESI) models [70], and capability maturity models (CMM) for companies [117]. When using indicator models for evaluation, research has been conducted from a variety of perspectives, including disaster-risk management [118], community-resilience assessment in specific regions [109,119,120] and urban land-ecosystem-resilience evaluation [121,122]. Depending on the topic and perspective, these models can be developed and applied to flood resilience studies in different ways. Model-based research on urban flood resilience will not only play a huge role in urban risk assessment in the future, but also improve the accuracy of assessment. In addition, it is an indispensable and important research method to study urban flood resistance and vulnerability.

## 4. Conclusions

As an important basis of an urban system’s ability to withstand rainstorm disasters, urban flood resilience is an important index of urban planning and management for the future. With the development of research in various fields, there has been an explosion of research on urban flood resilience in the water industry since 2016, and it has been expanding both theoretically and empirically. In terms of research content, the research focuses on climate change adaptation, disaster resilience, urban planning and management, and urban risk tolerance. The research methodological relationship gradually shifts from economic data analysis based on the analysis of macroscopic phenomena to the trend of obtaining more accurate evaluation results. Therefore, future research can be conducted in depth on the following aspects of the subject:Scientometric analysis provides a clear and quantitatively explicit presentation of results in the research literature related to urban flood resilience. Methods such as the co-citation analysis of keywords, authors, countries and research institutions are important for sorting out the changing research focuses and the development of research methods on urban flood resilience. The results of keyword co-occurrence analysis and cluster analysis are the basis for systematic analysis. The keywords and other keywords in urban flood restoration literature are quantitatively analyzed by scientists, and the research emphasis, regional distribution, research trends and change in research hotspots of relevant research literature can be displayed by specific numerical values. Keywords with a high frequency of occurrence, such as climate change, resilience research and other keywords, lay a foundation for the selection and classification of fields for subsequent systematic analysis. Through the literature analysis of relevant research institutions, it is possible to select highly cited and hot literature quickly. The quantitative analysis of keywords and other keywords in the literature related to urban flood resilience using scientists’ quantitative analysis can show the research focus, research area distribution, research trends and research hotspot changes of related research literature separately with specific values. Over the researched time span, urban flood resilience has concerned scholars from several regions and research institutions. There have been certain studies on risk assessment and resilience index assessment, but the research on models needs to be more comprehensive. The research fervor in recent years is in a stage of enhancement. In terms of the number of studies, urban flood resilience studies at this stage are still gradually increasing and are in a phase of yearly growth. Scientometric analysis is very effective for the analysis of literature related to this research topic, which can increase the accuracy of the analysis results and help a lot for subsequent research.Systematic analysis of the results of urban flood resilience and adaptation to climate change is a hot research area in urban flood resilience. It covers studies related to urban resilience to climate change, which are generally large in scale, and some of them include areas at a county level, while the specialized studies for communities are not comprehensive enough. Many of them use indicator evaluation models. The combination of higher performance and faster computing and simulation models will achieve the effect of refined research, which will lead to more accurate and applicable recommendations for urban development. Globally, urban flood resilience focuses more on carrying capacity and post-disaster recovery, while urban flood resilience requires more attention to the stability and resilience of urban structures in response to disasters, with little research on post-flood urban ecology. Urban flood resilience is closely related to urban planning, which mainly includes the influence of two main aspects: urban planning and design and the urban drainage system. Additionally, study is inseparable from the local country’s conditions and policies. The study of urban flood resilience based on changes in local conditions is also used as an important tool to adapt to the trend of the times and respond to current changes. A more fine-grained and high-precision assessment of urban flood resilience driven by multiple factors based on economic development, infrastructure, ecological conditions, urban drainage planning, and other urban conditions at moderately different levels will be a future development direction. At the same time, it is necessary to strengthen the refined research on flood readaptation at the large-scale level and expand the research methods at the small-scale level, then further improve the resilience assessment methods to better describe urban flood resilience.The relationship between urban resilience and urban-system resilience regarding disaster analysis is noteworthy. In recent years, the analysis and research on urban flood resilience using a combination of scientometric and systematic approaches can better show hotspots. The current research on urban flood resilience focuses on the relationship between climate change, urban infrastructure, urban management and environmental vulnerability, applied models and other factors. However, there are various analytical methods and evaluation indicators available for the analysis of urban flood resilience. More appropriate models and indicators can be selected for analysis to improve urban flood resilience, depending on the variability of the study area. Scientometric analysis can increase cross-sectional comparisons and enable further expansion of the scope of the research topic. At the same time, systematic scientometric research can progress research according to different thematic classifications, allowing for more detailed analysis of the literature. Thus, urban flooding issues based on a combination of scientometric and systematic analysis can be closely integrated with practical urban issues and policies, such as urban green infrastructure, urban water regulation and community differences.

## Figures and Tables

**Figure 1 ijerph-19-08837-f001:**
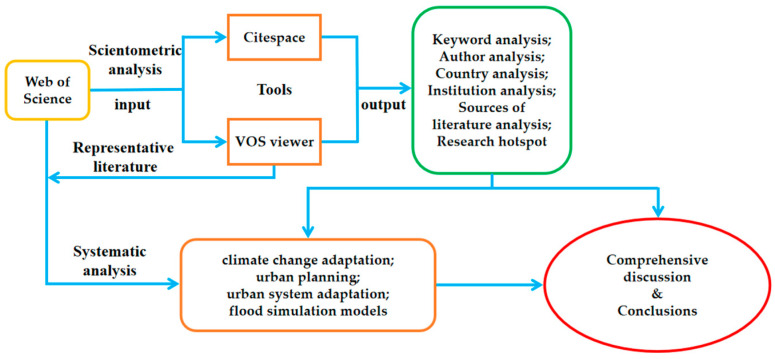
Process diagram of literature analysis.

**Figure 2 ijerph-19-08837-f002:**
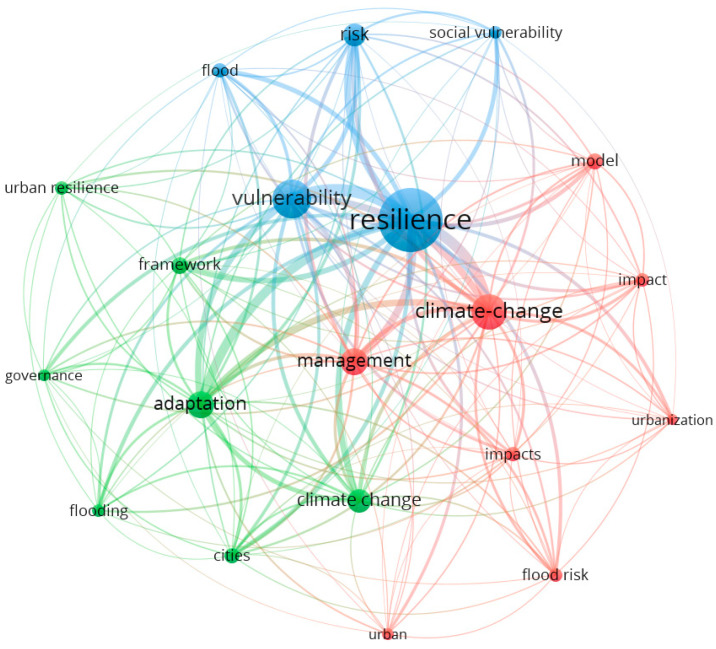
Keyword view in domestic urban flood-resilience-related literature from 1999 to 2021.

**Figure 3 ijerph-19-08837-f003:**
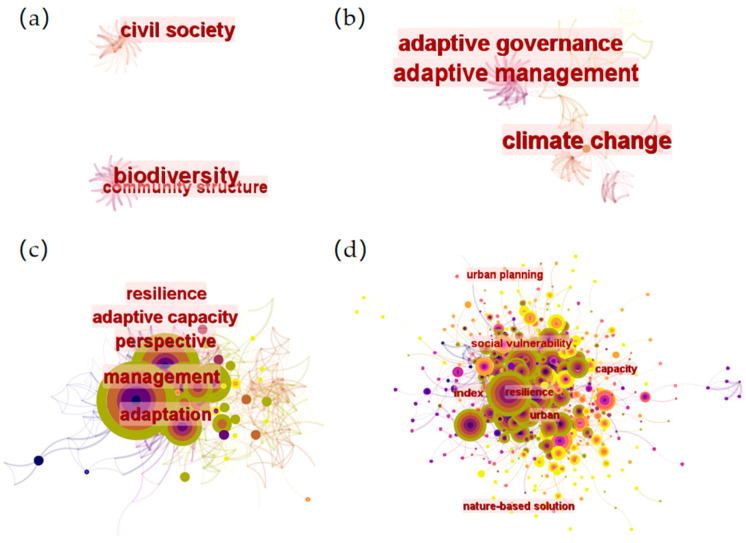
(**a**) Keyword analysis from 1999 to 2005; (**b**) from 2006 to 2010; (**c**) from 2011 to 2015; and (**d**) from 2016 to 2021.

**Figure 4 ijerph-19-08837-f004:**
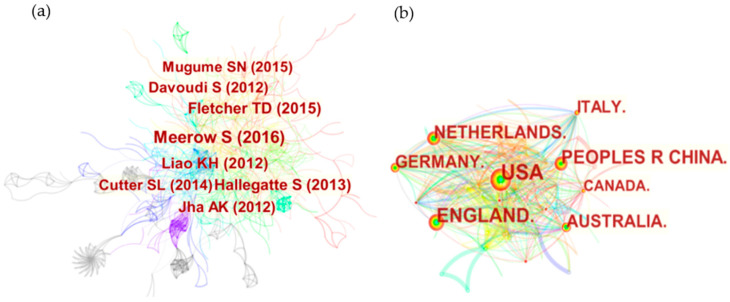
(**a**) Author-co-citation analysis from 1999 to 2021; (**b**) National-cooperative analysis from 1999 to 2021.

**Figure 5 ijerph-19-08837-f005:**
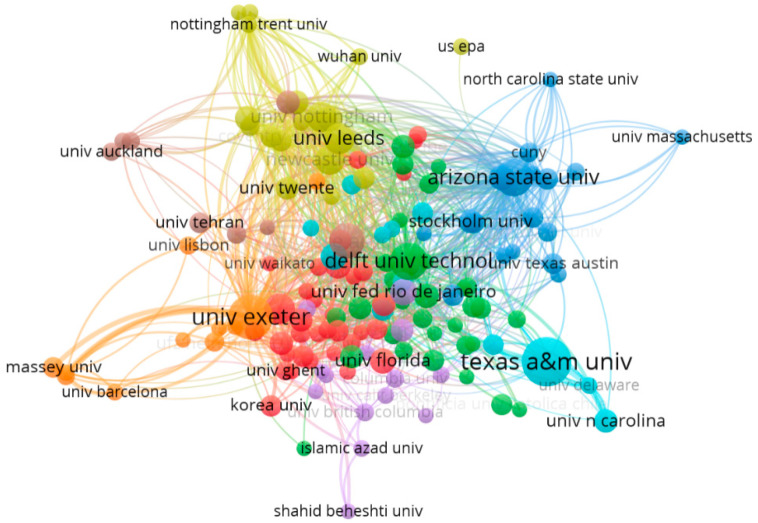
Network of structural relationships for the study of urban flood resilience 1999–2021.

**Figure 6 ijerph-19-08837-f006:**
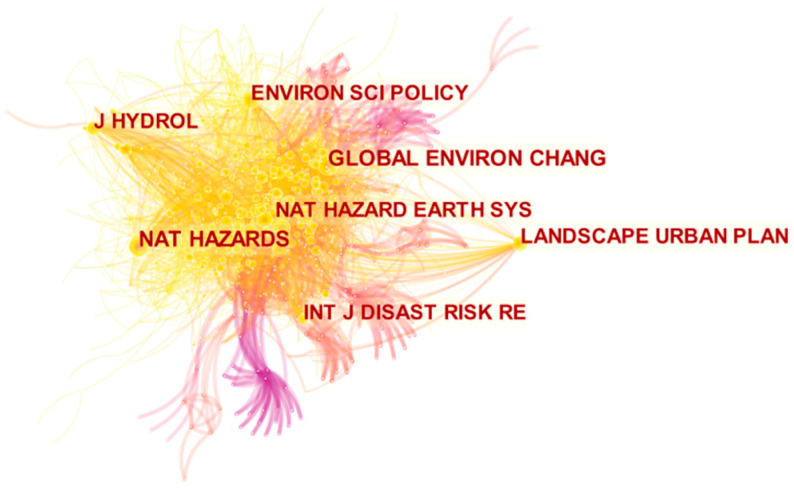
Journal cooperative network of the study of urban flood resilience 1999–2021.

**Figure 7 ijerph-19-08837-f007:**
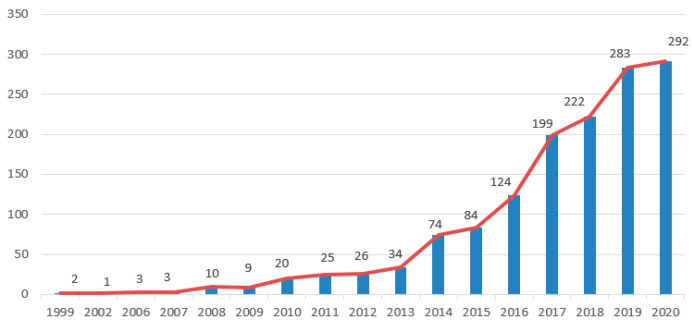
Increasing trend of literature about urban flood resilience.

**Table 1 ijerph-19-08837-t001:** Table of the top ten number of papers published by research organizations.

No.	Organizations	Number of Literatures on Urban Flood Resilience	Citations
1	Texas A & M University	52	514
2	University of Exeter	42	1067
3	Arizona State University	30	732
4	Delft University of Technology	30	734
5	Chinese Academy of Sciences	27	299
6	University of Leeds	20	245
7	University of Nottingham	19	301
8	Newcastle University	18	261
9	Portland State University	14	325
10	Stockholm University	12	250

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
