# Peer review of "Review of Urban Flood Resilience: Insights from Scientometric and Systematic Analysis"

_ijerph, 2022, doi:10.3390/ijerph19148837_

Round 1
Reviewer 1 Report
1- Since the readers of this article are international, it is necessary to use international units. For example, instead of the yuan as a currency, it is suggested to use the dollar.
2- In line 56 – 61 reference should be provided.
3- The article deals with the destructive effects of floods in urban environments. Look out of this window that urban floods can help the process of feeding groundwater resources and even prevent land use change in rivers at urban areas and can even cause the transfer of waste products. Related articles should also be considered and included in the article.
4- In the material and method section, it is better to draw the process in the form of a figure and add it to the article.
5- In line 161, I do not know what you mean by this sentences, " A total of 146 keywords were analyzed and 161 497 keywords were connected".
6- In line 164, It is necessary to express the number of each category.
7- In Figure 1 -5, enter the legend and scale. Also what does the color spectrum represent? The quality of Figure 2 is not appropriate; a more appropriate figure should be provided.
8- Figure 2 shows the keywords and their variations. Is it possible to provide this information for geographical distributions as well? Have these keywords changed in different parts of the world? What could be the reason for these changes in keyword structure?
9- The information provided in Figure 4 should also be presented in table form to determine the rankings of universities. The names of some universities are not well illustrated.
Reviewer 2 Report
General Comments:
This study is a type of cluster analysis reviewing the type and quantity of papers that study urban flooding and climate change. Both of these are important topics these days. As a review paper, one would expect this to compile what is known about the connection between these two topics in the literature. They studied this by looking at key words, countries, authors, and journals. They look at the interconnections between these based on journal articles. The article is well referenced. The procedures used are done correctly. This paper does not need major revision, but the language does need revision throughout. It would be helpful to get assistance from a native English speaker. Also, be careful about capitalizing proper names, for example on line 312, "ornstein-uhlenbeck process" should be capitalized. Check the manuscript throughout.
Reviewer 3 Report
The paper needs major writing style revision and can also be improved as follows:
- remove inappropriate repetition of the same concept in different ways
- add a definition of urban flood resilience and its importance
- explain how the stages in line 177 are selected
- explain the results and discuss the figures more
- lines 188-190: need a better explanation.
- line 198: what do you mean by distance?
- line 275; when you say resilience here is it only for floods or in general?
-line 282: which year?
- line 289: explain why?
-line 412: there are different definitions of flood resiliency in the literature. perhaps the authors can discuss it and explain why they selected a certain definition.
- line 433: What is "Metri"? Do you mean Metrics?
- the conclusion does not reflect the paper review part
Round 2
Reviewer 1 Report
All the corrections made are approved.
Reviewer 3 Report
The authors have responded to the questions properly and improved the paper according to the comments.
Thank you